# Visceral Surgery Profoundly Affects the Cellular and Humoral Components of the Anti-Tumour Immune Response in a Murine Pancreatic Adenocarcinoma Model

**DOI:** 10.3390/cancers14163850

**Published:** 2022-08-09

**Authors:** Friederike Loening, Annabel Kleinwort, Lars Ivo Partecke, Tobias Schulze, Pia Menges

**Affiliations:** 1Division of General Surgery, Visceral, Thoracic and Vascular Surgery, Department of Surgery, University Medicine Greifswald, 17491 Greifswald, Germany; 2Department of General, Visceral and Thoracic Surgery, Helios Clinic Schleswig, 24837 Schleswig, Germany

**Keywords:** pancreatic ductal adenocarcinoma, surgically-induced immune dysfunction, murine pancreatic carcinoma model, anti-tumour immune response, immunotherapy

## Abstract

**Simple Summary:**

Surgery is a fundamental part of the multimodal therapy concepts in oncological patients, especially in the early stage of pancreas tumour. There are numerous studies on the impact of primary tumour resection on the immune status, but to our knowledge, the impact of tumour-unrelated surgery on the anti-tumour immune response to the primary tumour it is not yet understood. Therefore, we used a murine model combining an orthotopically implanted adenocarcinoma of the pancreas and the model of surgically-induced immune dysfunction to assess the impact of postoperative immunosuppression on the growth of the primary tumour, on mortality and on the most important immune cell compartments in tumour defence. This knowledge might contribute to a basic understanding of the interaction of the primary tumour and the immune system and could guide new approaches to therapeutic strategies.

**Abstract:**

(1) Background: Surgery is the most important element of multimodal treatment concepts in oncological patients, especially in the early stages of pancreatic tumours. While the influence of primary tumour resection on the immune status was analysed in several studies, the impact of tumour-unrelated visceral surgery on the tumour-bearing organism and on the primary tumour itself is not yet fully understood. (2) Methods: We combined a murine model of orthotopically implanted adenocarcinoma of the pancreas with the model of surgically-induced immune dysfunction (SID). Mortality and general condition including body weight were observed over a period of 28 days. Tumour growth was analysed by MRI scans on days 8 and 27 following tumour implantation. On day 28, the immune cell populations in the blood and spleen as well as the serum cytokines were quantified. (3) Results: SID results in a significant deterioration of the general condition and a reduced increase in the body weight of tumour-bearing mice compared to the control groups, while mortality and tumour growth rate were not influenced. The numbers of spleen macrophages and neutrophils were increased in tumour-bearing animals following SID. Furthermore, both macrophage and neutrophil levels were increased in the peripheral blood. (4) Conclusions: The presented results might contribute to the basic understanding of the interaction of tumour and immune system and could contribute to new approaches to immunotherapeutic strategies.

## 1. Introduction

Surgical procedures induce a state of immunosuppression for up to several weeks postoperatively. The immune dysfunction affects both the innate and the acquired immune system [1]. The innate immune system represents the first line of immune defence against invading pathogens, participates in wound healing, and is involved in regulation of the antitumour immune response as well as tumourigenesis [2,3,4]. It interacts closely with and activates the adaptive immune response. Specific anti-infectious and anti-tumour immune reactions are linked to the activities of cytotoxic T-cells, which are subjected to various inhibitory signals (regulatory T-cells, IDO expression, and expression of inhibitory surface molecules) within the tumour microenvironment [2,5]. Postoperative immunosuppression is the consequence of the activation of the hypothalamic–pituitary–adrenal axis and is mediated by the effects of corticosteroids and catecholamines on the cells of both the innate and the adaptive immune system (for review, see [1]). The clinical consequences of this surgery induced immunosuppression (SII) comprising an increased incidence of postoperative local and systemic infections, disturbances of wound healing, and a deterioration of oncological prognosis in the case of tumour-associated surgery [6,7,8,9,10,11].

Surgery is a very important element of multimodal treatment concepts in oncological patients. However, the surgical procedure and the associated stressors have also been shown to increase the susceptibility for the development of cancer metastasis and local recurrence (for review, see [12]). Several mechanisms have been implicated in this phenomenon. Among them, SII leads to the suppression of cellular immunity with the consequence of an immune escape of dormant or clinical unapparent tumour cells [13,14]. However, the removal of the primary tumour itself also seems to influence the development of local and distant recurrence by causing a drop in the levels of tumour-related anti-angiogenic factors (e.g., angiostatin and endostatin) [15,16].

Most of the in vivo and in vitro studies have evaluated the effects of SII on cancer recurrence and the development of metastasis in the context of the resection of the primary tumour. Since primary tumours have been shown to supress the development of metastasis (for review, see [17]), it is difficult to differentiate between the effects of perioperative immunosuppression and the effects of the resection of the primary tumour in this kind of model. Moreover, the metastatic immune microenvironment has been shown to be markedly different from that of the primary tumour [18]. In the present work, the effect of non-tumour related surgery on the growth and prognosis of pancreatic carcinoma was assessed in a murine model combining an orthotropic pancreatic cancer model [19] and the model of surgery induced immunosuppression (SID) [20]. The single and combined effects of the presence of the primary tumour and the surgical trauma on the immune system were extensively characterised in this combined model in order to better understand the respective importance of both phenomena.

## 2. Materials and Methods

*Animals.* Nine week old male C57BL/6N mice purchased from Charles River Laboratories (Sulzfeld, Germany) were held under specific pathogen-free conditions at least two weeks before inclusion into the experiments in order to adapt to the housing conditions. Mice had free access to food and water. The animal experiments were approved by the veterinary government authority (Landesamt für Landwirtschaft, Lebensmittelsicherheit und Fischerei in Mecklenburg-Vorpommern (LALLF M-V); experiment numbers: TSD/7221.3-1-038/16 and TSD/7221.3-1-038/16-1). Anaesthesia for tumour implantation, operations, and organ removal was performed by intraperitoneal injection of 100 mg/kg body weight ketamine (Selectavet Dr. Otto Fischer GmbH, Weyarn-Holzolling, Germany) and 10 mg/kg body weight xylazine (Bayer, Leverkusen, Germany). Anaesthesia for MRI examination was performed by inhalational anaesthesia with 1% isoflurane. After interventions, drinking water was supplemented with 250 mg/100 mL tramadole (Grünenthal, Aachen, Germany). At the end of the experiment or when the termination criteria of the stress score were fulfilled, mice were euthanised under intraperitoneal injection narcosis with ketamine and xylazine by cervical dislocation.

*Cell line.* The murine pancreatic adenocarcinoma cell line 6606PDA was a kind gift from Prof. David Tuveson, Cambridge, UK (current address: Cold Spring Harbour Laboratories, New York, NY, USA) and was isolated from pancreatic adenocarcinoma cells in transgenic C57BL6/N mice carrying a KrasD12G allele [21].

*Experiment design.* Animals were randomly divided into six experimental groups: SID + tumour, laparotomy + tumour, without operation (w/o op) + tumour, SID, laparotomy, and the control. On day 0, 6606-PDA-cells were xenografted in mice by orthotopic injection into the pancreas as described before [19]. In brief, 2.5 × 10^5^ pancreatic carcinoma cells dissolved in PBS (PAN™ Biotech, Aidenbach, Germany) and Matrigel (BD Bioscience, Belgium) were injected into the head of the pancreas. On days 8 and 27, the tumour volume was determined by magnetic resonance imaging using a 7.1 tesla ClinScan device (Bruker, Bremen, Germany), generating T2-weighted axial and coronal images. Images were analyses with the MIPAV software (Medical Image Processing, Analysis, and Visualisation, independent Java application). On day 11 and on day 21, surgery was performed (Figure 1). Animals underwent either SID, a median laparotomy, or no operation at all. For SID, median laparotomy was performed, and the small intestine was gently squeezed between two moistened cotton swabs in an antegrade direction for three times as described in detail by Koerner et al. [20].

*Stress score.* The animals were assessed daily over the complete experimental duration by determining a stress score and the body weight. The stress score was designed for mice suffering from pancreatic tumour disease, as described by Partecke et al. [22] and includes the following items: physical appearance (normal fur = 0; slightly ruffled fur = 1; moist, ruffled fur = 2; moist, ruffled fur with mucous eye discharge = 3); respiration (normal = 0; tachypnoea = 1; dyspnoea = 2; shallow breathing = 3); spontaneous behaviour (normal = 0; reduced, slowed locomotion = 1; unsteady gait, kyphotic posture = 2; lateral position = 3) and elicited behaviour (normal = 0; escape behaviour triggered by physical approach = 1; escape behaviour triggered by physical contact = 2; no escape behaviour = 3). The total clinical score was obtained by the addition of the item scores for each time point. When reaching a score ≥7, animals were excluded from the experiment and euthanised as described above.

*Blood cell counts.* On day 28, the animals were narcotised as described above and blood samples were taken by retrobulbar blood collection. Blood cell counts were analysed by a veterinary haematology analyser (VetScan HM5™, Abraxis, Warminster, PA, USA).

*Spleen cell isolation.* After cervical dislocation, the spleen was removed and stored on ice in PBS with 10% FCS (foetal calf serum, Biochrom GmbH, Berlin, Germany) until further processing. The spleen was dispersed by squeezing the organ through a 70 µm cell strainer (Becton Dickinson GmbH Heidelberg, Heidelberg, Germany) and erythrocytes were eliminated by a red blood cell lysis buffer (BioLegend, San Diego, CA, USA). Absolute cell numbers were determined using BD™ Trucount tubes (BD Biosciences, San Jose, CA, USA).

*Flow cytometry.* Nonspecific binding was blocked with an anti-FcγIII/II antibody (anti-CD16/32; BioLegend, San Diego, CA, USA). The following antibodies and conjugates were used in the experiments: anti-Ly6G-Alexa647 (clone 1b8) (BioLegend, San Diego, CA, USA), anti-CD3e-V500 (clone 500A2) (BD Biosciences, San Jose, CA, USA), anti-CD19-V450 (clone 1D3) (BD Biosciences, San Jose, CA, USA), anti-CD45-allophycocyanin (APC)/Cy7 (clone 30f11) (BD Biosciences, San Jose, CA, USA), anti-CD4-Alexa700 (clone RM4-5) (BioLegend, San Diego, CA, USA), anti-CD8-phycoerytrhin (PE)/Cy7 (clone 53-6.7) (BioLegend, San Diego, CA, USA), CD11b-brilliant violet (BV)650 (clone M1/70) (BioLegend, San Diego, CA, USA), and anti-NKp46-PE (clone 29A1.4.) (Miltenyi Biotec, Bergisch Gladbach, Germany). For dead cell exclusion, the cells were stained with 7-AAD Viability Staining Solution (Biolegend, San Diego, CA, USA) before analysis. Stained cells were analysed on a BD LSR II Flow Cytometer (BD Biosciences, San Jose, CA, USA) and evaluated with FlowJo software (Version 10, Tree Star Inc., Ashland, OR, USA). Total spleen immune cells were defined as 7AAD^−^CD45^+^ cells, macrophages as 7AAD^−^CD45^+^CD11b^+^Ly6G^−^, neutrophils as 7AAD^−^CD45^+^CD11b^+^Ly6G^+^, B-cells as 7AAD^−^CD45^+^CD19^+^, T-cells as 7AAD^−^CD45^+^CD3^+^, and NK-cells as 7AAD^−^CD45^+^NKp46^+^ cells. T-cells were subdivided into T-helper cells (CD4^+^) and cytotoxic T-cells (CD8^+^).

*Cytokine analysis.* Cell suspension from the spleen was incubated with RPMI (Gibco by Life Technologies, Waltham, MA, USA) + 10% FCS + 1% penicillin/streptomycin (Gibco by Life Technologies, Waltham, MA, USA) for 24 h at 37 °C and 5% CO_2_. The serum of 100 µL heparinised blood was frozen at −20 °C until further analysis. Serum and supernatants of the spleen were analysed by the Cytometric Bead Array (CBA) using the BD™ CBA Mouse/Rat Soluble Protein Master Buffer Kit and BD™ CBA Mouse Flex Sets (BD Biosciences, San Jose, CA, USA) according to the manufacturer’s protocol. The following cytokines were analysed: IL-6, IL-10, TNF-α, INF-γ, IL-4, IL-1β, IL-6, IL-13, MCP-1, and IL-21. Stained beads were analysed on a BD LSR II Flow Cytometer (BD Biosciences, San Jose, CA, USA) and evaluated with the FCAP Array™ software (Soft Flow, Pécs, Hungary).

*Statistical analysis.* Statistical analysis and graphs were performed with the GraphPad Prism software (GraphPad Software Inc., San Diego, CA, USA). The groups were tested for Gaussian distribution with the Shapiro–Wilk test. The three tumour-bearing operation groups were compared to its corresponding non-tumour-bearing group by the Student’s t-test when they were normally distributed; otherwise, the Mann–Whitney–U test was used. For multiple comparisons within the tumour-bearing groups (and control) or within the non-tumour-bearing groups, a one-way ANOVA, when normally distributed, or a Kruskal–Wallis test was performed. The survival rate was determined by Kaplan–Meier curve. *p* values < 0.05 were considered significant (* *p* < 0.05 and ** *p* < 0.01). The graphs show the mean and standard error when normally distributed and the median and interquartile ranges otherwise.

## 3. Results

### 3.1. SID Results in a Significant Deterioration of the General Condition of Tumour-Bearing Mice

The general condition of mice included in the experiments was assessed using the stress score modified from Partecke et al. [22]. As expected, tumour-bearing mice had higher stress scores than tumour-free animals (Figure 2a). Among the tumour-bearing mice, animals subjected to SID presented with the highest stress scores, compared to animals after laparotomy only and mice undergoing no surgery at all (Figure 2b).

### 3.2. Impact of SID on Weight Development

While the body weight of the control animals increased by 10.4% during the observation period, the tumour-bearing animals showed significantly smaller increases in body weight. Among the tumour-bearing mice, those undergoing SID showed a tendency to increase their body weight to the smallest extent among all of the experimental groups observed (Figure 3).

### 3.3. Growth of the Primary Tumour Was Not Influenced by SID

In order to assess whether SID resulted in increased growth of the primary tumour, the growth of the orthotopic pancreatic tumour was assessed by MRT scan on day 8 (8 days after tumour implantation and 3 days before SID) and day 27 (after two episodes of SID). Tumour growth was similar in animals with no surgical trauma, minor trauma (laparotomy), and SID (Figure 4).

### 3.4. No Impact of SID on the Survival Kinetics in Tumour-Bearing Mice

Visceral surgery unrelated to the primary tumour had no impact on the survival of tumour-bearing mice. The survival in tumour-bearing animals was similar in animals subjected to limited (laparotomy, 75%) or extensive (SID, 85%) surgical trauma or no surgical trauma at all (73%) (Figure 5). Tumour-free mice without surgery or subjected to laparotomy showed no mortality, in contrast to tumour-free mice subjected to SID, which showed a survival of only 78% (Figure 5). These data indicate that surgery not related to the primary tumour had no impact on the tumour-associated mortality in the murine orthotopic pancreatic tumour model.

### 3.5. SID Results in a Profound Modification of the Composition of Blood Immune Cell Populations in the Blood of Tumour-Bearing Mice

The total leucocyte counts were increased in tumour-bearing animals subjected to SID even 7 days after the last surgical procedure. In contrast, tumour-bearing mice without intercurrent surgery or with only minor surgical trauma showed no increase in the total leucocyte numbers compared to the healthy controls (Figure 6a). This difference was due to a significant increase in the monocyte and neutrophil numbers (Figure 6b,c), while the lymphocytes showed a tendency to lower numbers in animals with intercurrent SID or laparotomy compared to tumour-bearing mice without intercurrent surgery or healthy controls (Figure 6d). The thrombocyte counts were also increased in tumour-bearing animals with intercurrent SID compared to the healthy controls (Figure 6e).

### 3.6. Immune Cell Populations in the Spleen Show Significant Quantitative Alterations after Severe Surgical Trauma

The absolute spleen cell numbers were significantly increased in the tumour-bearing mice subjected to intercurrent SID compared to healthy control animals, while no significant increase could be detected between the later and tumour-bearing animals after intercurrent laparotomy or after no intercurrent surgery (Figure 7a). This observation was mainly due to increased macrophage and neutrophil numbers (Figure 7b,c), while the absolute numbers of splenic B- and T-cells as well as NK-cells showed no significant differences (Figure A1a–c). Interestingly, the absolute splenic macrophage numbers were increased in all groups of tumour-bearing animals compared to the healthy control animals (Figure 7b).

The comparison of relative cell numbers showed a significant decrease in both the splenic B- and T-cells in tumour-bearing SID animals compared to the healthy controls, but also to tumour-bearing animals without intercurrent surgery (Figure 7d,e). Both splenic macrophages and neutrophils were increased in tumour-bearing SID animals compared to the healthy controls, but also to tumour-bearing animals without intercurrent surgery (Figure 7f,g). The differences in B-cell, T-cell, neutrophil, and macrophage numbers were also present in the comparison of tumour-free control animals and tumour- free animals subjected to laparotomy and SID (Figure A1d–g). NK-cell relative cell numbers were decreased in the spleen of all tumour-bearing animals compared to the tumour-free control animals (Figure 7h).

A detailed analysis of the T-cell populations revealed a significant decrease in both the CD4^+^ T-helper cells and CD8^+^ cytotoxic T-cells in the tumour-bearing SID animals compared to the tumour-bearing animals without operations as well as the tumour-free control animals (Figure 7i,j).

### 3.7. Both Pro- and Anti-Inflammatory Cytokines Are Increased after Non-Oncological Surgery in Tumour-Bearing Mice

The plasma concentrations of IL-6, IL-10, and TNF-α were significantly increased in the serum of tumour-bearing SID animals compared to the healthy controls (Figure 8a–c). Both IL-6 and TNF-α were also increased in the tumour-bearing SID mice compared to the tumour-bearing mice without intercurrent surgery. The IL-1β, MCP-1, and IL-4 levels showed no significant differences between the groups (Figure A2a–c). The levels of IFNγ, IL-12p70, IL-13, and IL-21 were not detectable under the experimental conditions used (Figure A2d–g).

In spleen cell supernatants, IFN-γ, IL-4, IL-1β, IL-6, IL-13, MCP-1, and IL-21 were significantly higher in the tumour-bearing SID animals compared to the healthy controls (Figure 8d–j). The IL-2, IL-10, and TNF-α levels showed no significant differences (Figure A3).

## 4. Discussion

Pancreatic ductal adenocarcinomas (PDAC) are characterized by the capability to exercise a profound modification and inhibition of both the local and systemic immune response [23]. Thus, surgical procedures aimed at the removal of the primary tumour impact the immune status not only by mechanisms linked to the SII, but also by the consecutive lack of immunomodulating signals produced by the primary tumour. Therefore, we used a murine model combining an orthotopically implanted PDAC and SID to assess the impact of SII on the growth of the primary tumour itself as well as survival. The results presented in this paper clearly show that SII induced by visceral surgery not related to the primary tumour does not result in an increased tumour growth or reduced overall-survival. However, there was a tendency to reduced weight gain of tumour-bearing animals and the general condition was severely impaired by SII.

In the present study, SII had no impact on the overall survival. This is in contrast to the results reported by Menges et al. [24]. In this study, tumour-bearing animals subjected to SID showed a significantly reduced survival compared to the non-operated control mice, while tumour growth was similar in both experimental groups. In contrast to the present study, the follow-up of the survival analysis was significantly longer with 60 days post-implantation of the PDAC. This difference in the length of the observation period may account for the differences observed between the present study and the report of Menges et al., as the main proportion of mortality in the groups analysed in that study was observed between day 20 and day 60, while the present study was designed to evaluate mortality for a period of only 28 days.

Our results indicate that SID leads to a decline in the general condition of the tumour-bearing mice, which is supported by the tendency to a reduced weight gain in the period of 28 days following the tumour implantation as well as a significantly increased stress score. It is well-established that surgical stress provokes a physiological response with the release of glucocorticoids and the neurotransmitters epinephrine and norepinephrine in a multifactorial context [25]. The resulting alterations of metabolic processes including protein catabolism are known to have a direct influence on the weight development in mice [26]. This agrees with our observation of a tendency to reduced weight gain in the group of tumour-bearing mice undergoing SID, whereas their organism is already characterized by a catabolic status due to the present tumour.

In accordance with previous studies, we observed no influence of SID on the tumour growth as analysed by MRI scan on day 8 and day 27 following the tumour implantation [24]. The increase in tumour volume is only one aspect of the disease progress, while the interaction of immune response and tumour progress is most complex. SII might therefore result in alterations on the cell levels and could change the tumour microenvironment (TME), but does not influence tumour growth in the first line. Further studies to prove this hypothesis are considered.

The tumour generated by the 6606PDA cell line is considered to be valuable in characterising several aspects of cancer immunology: inflammatory cells are shown to surround but not to infiltrate the carcinoma [27]. The tumour based on this cell line is further known to represent a moderately differentiated glandular tumour growth [19]. The clinical relevance of this tumour model was proofed by Partecke et al. [28]. We therefore conclude that our herein presented model may be useful to analyse the impact of non-tumour related surgery on a “cold” tumour with a glandular growth pattern and thus in a widespread type of PDAC.

The immune microenvironment of PDAC has been characterised as immunosuppressive and up to now, immunotherapeutic approaches to this tumour entity have mostly shown unsatisfactory results [23,29]. This is mainly due to several immune escape mechanism of the pancreas tumour (i.e., the release of immune-suppressive chemokines or cytokines by the tumour and the establishment of a barrier of fibroblasts and collagen [30]). There are numerous clinical studies under progress that have different strategies in immunological treatments against PDAC [31]. Our presented murine model could be valuable in studying the influence of non-tumour related surgery on the effect of several immune therapeutic approaches.

A profound knowledge of the impact of visceral surgery, which is part of the multimodal treatments against PDAC, on the immune system of tumour-bearing animals is of importance. In our model, SII was characterised by changes in the immune system, mostly described to hamper the anti-tumour immune response.

Increased numbers of cytotoxic CD8+ T cells and T-helper 1 T-cells have been shown to mediate the anti-tumour response in murine models of PDAC. Increased numbers of these cell types are correlated with an improved survival in human PDAC patients [32]. In our model, both CD8^+^ and CD4^+^ T-cell populations had been reduced in tumour-bearing animals after SID, indicating that a reduced T-cell associated anti-tumour immune response induced by trauma due to surgery is not related to the primary tumour. Further characterisation of the tumour infiltrating T-cell populations in this model is required to address the local consequences of this observation.

Tumour-associated macrophages (TAM) and tumour-associated neutrophils (TAN) are essential components of the TME and have a profound impact on the biological behaviour of PDAC [23]. Increased numbers of M2 polarized TAM correlate with poor prognosis in PDAC patients [33]. Similarly, higher numbers of TAN in the TME correlate with poor survival in PDAC [34]. Interestingly, the spleen and not only the bone marrow has been shown to be an essential source of both TAM and TAN, which are constantly refurnished by the spleen during tumour progression [35]. Noteworthy, in the combined model used in this study, the numbers of spleen macrophages and neutrophils were increased in the tumour-bearing animals after intercurrent SID. Both macrophage and neutrophil numbers were also increased in the peripheral blood. These findings suggest that surgical trauma increases the splenic reservoir for macrophages and neutrophils that are capable of infiltrating the growing primary tumour. Further histological studies of the primary tumour and the spleen are required to verify this hypothesis. TGF-β has been shown to induce a pro-tumour phenotype in TAN [36]. In the model used in this work, TGF-β was significantly increased in the plasma of tumour-bearing mice after intercurrent SID. This suggests that SII both increases the number of potentially tumour infiltrating splenic neutrophils and contributes to a pro-tumour polarisation of these cells.

IL-4 and IL-13 have both been shown to supress the anti-tumoural immune response, and moreover to directly drive the proliferation of KRAS-mutant tumour cells [37]. The levels of both cytokines have been shown to be increased in the spleen of tumour-bearing mice after SID treatment. Given that the tumour growth was not different in the experimental groups of the trial presented herein, the clinical relevance of the observation remains unclear. Since the follow-up in our experimental setting was relatively short, this question should be addressed in an experimental setting with longer follow-up. IL-10, which is known to have tumour promoting effects, has been also increased in the serum of tumour-bearing mice after SID treatment [38]. These observations, taken together, suggest the presence of an internal milieu favouring the suppression of an anti-tumour immune response in tumour-bearing animals after SID.

In this paper, we describe the systemic consequences of surgical procedures on the systemic immune status in mice bearing a pancreatic tumour. However, further research based on the findings in the present manuscript is required to complete the general picture of an immunomodulating influence of non-tumour related abdominal surgery in tumour-bearing mice. First, a histopathological analysis of the primary tumour is required to assess the local consequences of the observed systemic changes of the immune system. In pancreatic cancer, both the tumour cells and the TME can exert local immunosuppressive actions through a wide range of different mechanisms (reviewed in [39]). Therefore, a detailed analysis of the immune cell infiltrate in the primary tumour as well as of its TME is needed to contextualise the systemic changes observed in our work. Moreover, a comprehensive phenotyping of the immune cell populations concerned by the changes described in this manuscript is necessary. Second, our model should be tested in an experimental setting with a prolonged observation period after SID, in order to determine whether the observed clinical and immunological consequences of SID in tumour-bearing animals will eventually result in differences in tumour growth and survival.

On the basis of the composition of the immune cell composition of the microenvironment, PDCA can be subdivided into distinct subtypes that are each associated with characteristic clinicopathological features and the presence of mutational changes in the tumour cells [40]. The mutational status of the malignant cells forming the PDCA seems to influence the development of the tumour microenvironment [40] and forms the basis for other molecular classifications of PDCA [41,42]. Although the mutational status of the orthotopically transplanted tumour cells in our experimental model was similar in the different experimental animals, it would be interesting to assess whether the modifications of the systemic immune status reported herein may impact the development of a specific PDCA phenotype over a longer follow-up period.

## 5. Conclusions

In summary, the results presented herein revealed a profound impact of surgery-associated trauma on the immune system of the tumour-bearing animal. Several of the modified immune parameters observed have been described to be of major importance in the host’s immunological anti-tumour response. Given that anti-tumour immunotherapy is a promising therapeutic modality in many solid tumours but has mostly failed so far in PDAC [29], surgery induced modifications of the immune status should be taken into account when new immunotherapeutic strategies are conceptualised in PDAC.

## Figures and Tables

**Figure 1 cancers-14-03850-f001:**
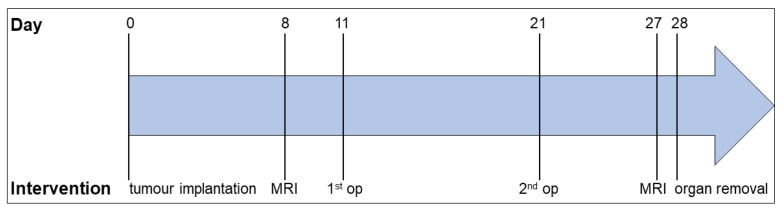
**Experimental design.** The experiment was set up for 28 days. On day 0, pancreatic tumour cells were implanted into the animal’s pancreas. On day 8, the tumour volume was determined by MRI. On days 11 and 21, SID/laparotomy/no operation was performed, followed by the second MRI scan on day 27. On day 28, the animals were euthanised, and the organs were removed for further analysis.

**Figure 2 cancers-14-03850-f002:**
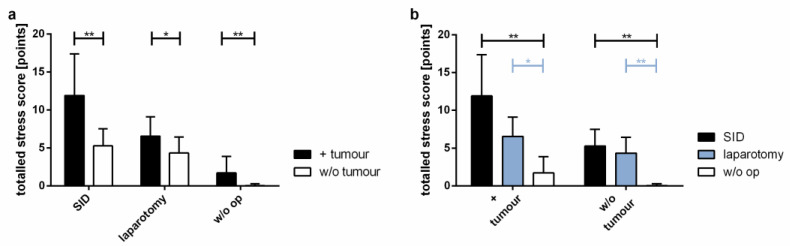
**Statistical evaluation of the stress level.** A stress score including physical appearance, respiration, spontaneous behaviour, and elicited behaviour was assessed every day to survey the influence of the tumour and/or the surgical procedure on the general condition. The graphic shows the stress scores of 28 days. (**a**) Comparison of tumour vs. w/o tumour in SID, laparotomy or in w/o operation; (**b**) comparison of SID vs. laparotomy vs. w/o operation in tumour-bearing mice and in tumour-free mice. Values represent the mean and standard error; *n* = 9–15 (SID + tumour: *n* = 11; laparotomy + tumour: 9; w/o op + tumour: 11; SID: 11; laparotomy/control: 15); ** p* < 0.05; *** p* < 0.01.

**Figure 3 cancers-14-03850-f003:**
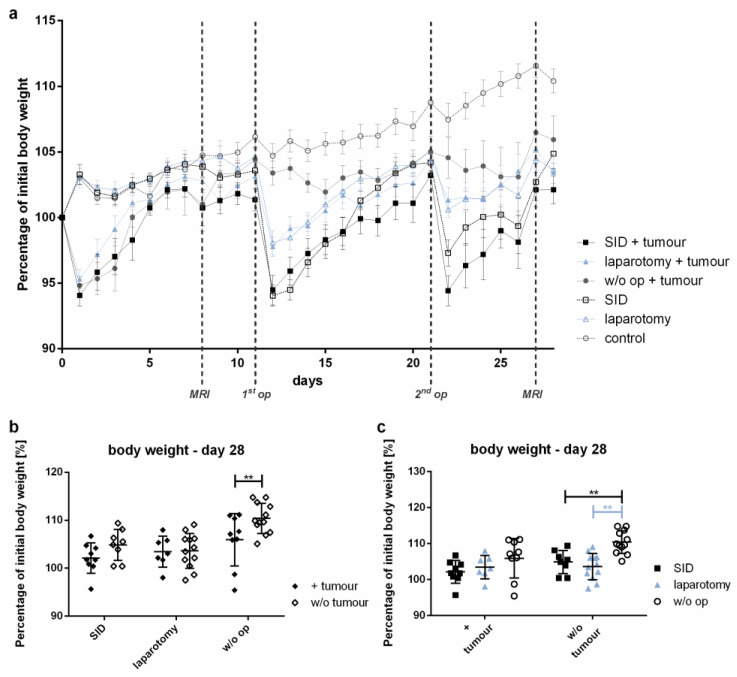
**Body weight development.** The body weight was determined daily. The body weight is shown as the percentage of the pre-interventional body weight (day 0). (**a**) The graph illustrates the body weight development during the experiment. (**b**) Comparison of the mean body weight at the end of the experiment (day 28) of the tumour vs. w/o tumour in SID, laparotomy or in w/o operation; (**c**) comparison of the mean body weight on day 28 of the SID vs. laparotomy vs. w/o operation in tumour-bearing mice and in tumour-free mice. (**a**) Values represent the mean and error of the mean; *n* = 15 animals per group; (**b**,**c**) values represent the mean and standard error, *n* = 9–11 (SID + tumour: *n* = 9; laparotomy + tumour: 7; w/o op + tumour: 9; SID: 8; laparotomy/control: 11); *** p* < 0.01.

**Figure 4 cancers-14-03850-f004:**
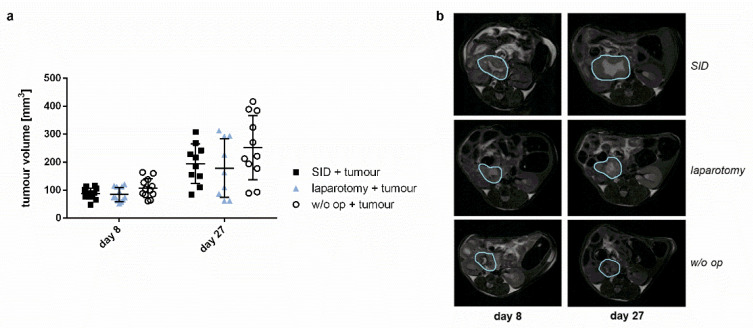
**Tumour volume.** Tumour volume was measured with MRI 8 days and 27 days after tumour implantation. There was no significant difference in the tumour growth between the operation groups. (**a**) Tumour volume 8 and 27 days after tumour implantation in mm^3^. Values represent mean and standard error; *n* = 9–12 (day 8: *n* = 12 for all groups; day 27: SID + tumour: *n* = 10, laparotomy + tumour: 9, w/o op + tumour: 11); (**b**) Representative MRI pictures of the tumour.

**Figure 5 cancers-14-03850-f005:**
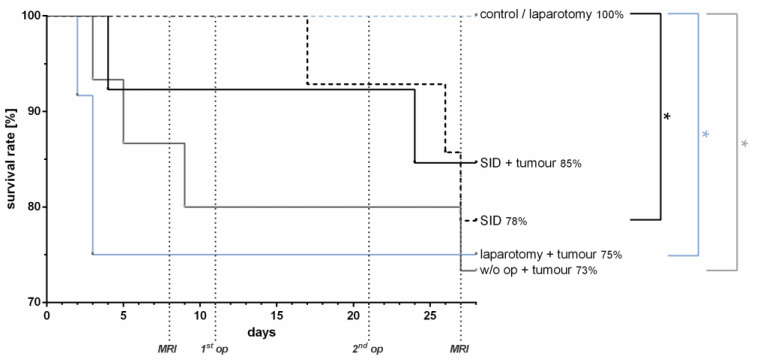
**Survival rate.** The survival rate was determined by the Kaplan–Meier curve and analysis by the Logrank test. The analysis showed no difference in survival rates between the tumour-bearing groups, whereas in the non-tumour-bearing groups, the survival was significantly reduced in the SID mice vs. control/laparotomy mice (* *p*). There was no significant difference in survival between SID with or without tumour, but between laparotomy + tumour vs. laparotomy (* *p*) and w/o op + tumour vs. control (* *p*). *n* = 12–15 (SID + tumour: *n* = 13; laparotomy + tumour: 12; w/o op + tumour/SID/laparotomy/control: 15); ** p* < 0.05.

**Figure 6 cancers-14-03850-f006:**
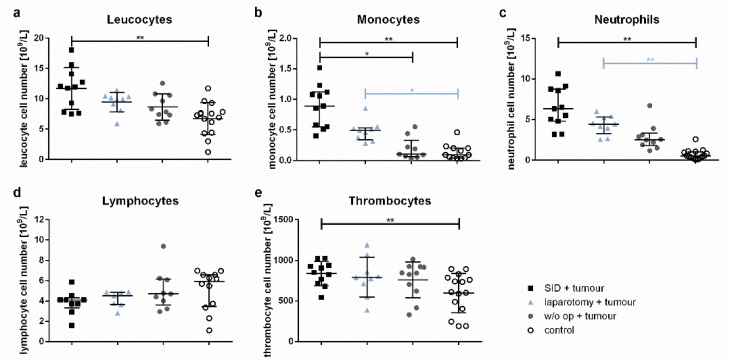
**Blood cell counts.** The blood cell counts of the venous blood samples were determined by a veterinary haematology analyser. The graphs show (**a**) the total leucocyte cell numbers, (**b**) monocyte, (**c**) neutrophil, (**d**) lymphocyte, and (**e**) thrombocyte cell numbers. Values in (**a**,**e**) are the mean and standard error; values in (**b**–**d**) represent the median and interquartile ranges with *n* = 9–15 animals per group (SID + tumour: *n* = 11; laparotomy + tumour: 9; w/o op + tumour: 11; control: 15); ** p* < 0.05; *** p* < 0.01.

**Figure 7 cancers-14-03850-f007:**
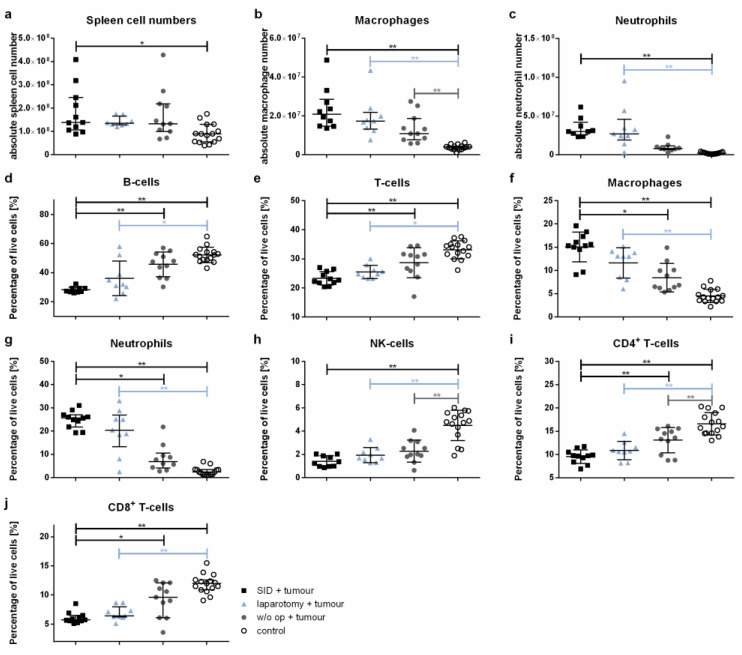
**Spleen cell populations.** The spleen cell populations were determined by flow cytometry. The graphics show either the absolute cell numbers or the percentage of vital spleen cells. Values in (**a**–**c**,**g**,**j**) represent the median with interquartile ranges; values in (**d**–**f**,**h**,**i**) are the mean and standard error with *n* = 9–15 animals per group (SID + tumour: *n* = 11; laparotomy + tumour: 9; w/o op + tumour: 11; control: 15); ** p* < 0.05; *** p* < 0.01.

**Figure 8 cancers-14-03850-f008:**
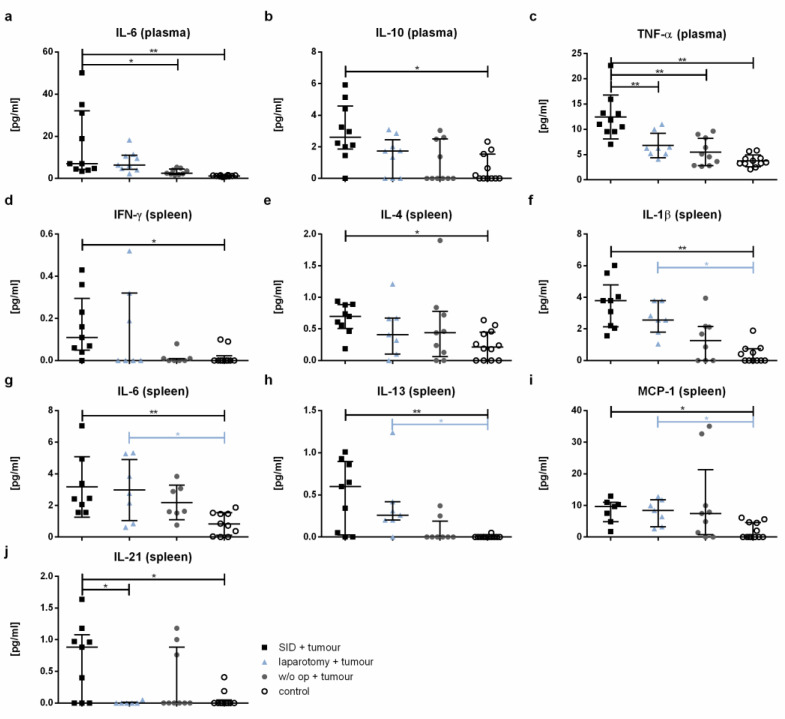
**Cytokine level.** The cytokine levels were measured by *CBA* in the spleen cell supernatant after 24 h incubation and in plasma. The graphics show the cytokine concentrations in pg/mL in either the plasma (**a**–**c**) or spleen cell supernatant (spleen; (**d**–**j**)). Values in (**a**,**b**,**d**–**f**,**h**–**j**) represent the median with interquartile ranges; values in (**c**,**g**) are the mean and standard error with *n* = 7–12 animals per group (SID + tumour: *n* = 9; laparotomy + tumour: 7; w/o op + tumour: 8; control: 12); ** p* < 0.05; *** p* < 0.01.

## Data Availability

Data are contained within the article.

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
