# Peer review of "Visceral Surgery Profoundly Affects the Cellular and Humoral Components of the Anti-Tumour Immune Response in a Murine Pancreatic Adenocarcinoma Model"

_cancers, 2022, doi:10.3390/cancers14163850_

Round 1
Reviewer 1 Report
The manuscript entitled "Visceral surgery profoundly affects the cellular and humoral components of the anti-tumour immune response in a murine pancreatic adenocarcinoma model" by Loening et. al endeavors to identify the role of postoperative immunosuppression in the absence of primary tumor resection. The authors found that SID did not affect the weight, survival or tumor growth in mice, but did diminish body condition and skew immune cells in the blood/spleen with a general increase in innate immune cells and decrease in adaptive populations. The manuscript is well-written and this concept is important/interesting, but I have some concerns regarding the data and conclusions of the paper.
Major concerns:
· Was onset of SID confirmed in comparison to laparotomy? Please show levels of IL-6/HMGB1 in the serum as described by Koern et. al. (6 or 24hr).
· The increase in macrophages/neutrophils and reduction in T/NK cells suggests that there could be reduced immune responses in the tumor, but this is impossible to discern with the data currently available. Does the drop in T cells/NK cells in these organs really mean there will be less antitumor activity, or could it be that these cells have different migration into the tumor itself, thereby affecting their total number in the blood/spleen? Would tumor growth/survival be different among the groups if the experiments went longer? Longer studies are required to interpret these findings and to better inform how non-tumour related surgery can affect the immune system.
· Typically, cytokines in the plasma are measured immediately and not following a 24-hour incubation. Did the authors look at cytokines directly from the blood without the 24-hour incubation? This rest period could skew the actual levels retrieved from the plasma. Splenocytes are typically activated prior to collection of supernatants for cytokines. Was there any activation of cells in vitro or were they just plated for 24 hours prior to collection? What was the reasoning behind this experimental design?
· While the changes in numbers of immune cells in the blood/spleen is intriguing, it's difficult to tie this to downstream changes in weight/tumor growth/etc. without knowing the phenotype of these cells and their presence in the tumor. Are the increased innate cells pro- or anti-inflammatory? The cytokines alone can't answer this question as there are both pro- and anti-inflammatory cytokines reported. Phenotyping these cells more in-depth and looking at the subsets in the tumor itself would help answer these questions.
Minor concerns:
· Please put figures in order of which they're introduced in the text (i.e. Figure 2A should come before Figure 2B in the text and figure).
· Figure numbering in the legends needs to be corrected to match the text.
· Please show available data instead of saying 'data not shown'
· Change 'Figure 8d-i' to 'Figure 8d-j'.
· Data is very bar graph-heavy. Please consider changing some of the figure displays to make it less monotonous.
· In the discussion, the authors state that "...weight gain of tumour-bearing animals was significantly reduced and the general condition was severely impaired by SII." This statement is a bit misleading. While SID did increase the stress score in tumour-bearing mice (Figure 2a), surgery did not significantly reduce the weight of tumour-bearing mice (Figure 3).
· Based on the stress score and weight graphs, SID or laparotomy seems to affect healthy mice to an even greater extent than tumour-bearing mice (compared to their respective controls). Can the authors speculate on why this might be happening?
Reviewer 2 Report
I have read with interest this manuscript investigating the relationship between surgery and immune-response in patients affected by pancreatic adenocarcinoma. The question is really interesting and this study may be a basis for future researches in this specific field. At the same time, this reviewer has some suggestion to improve this manuscript:
1) the authors should better discuss their findings and the limitation, taking into account that pancreatic ductal adenocarcinoma is universally considered as a "cold" tumor. However, some subtypes (e.g.; medullary, colloid,..) harboring microsatellite instability usually do show a marked inflammatory infiltration. This topic merit an adequate discussion; how can this model ricapitulate all possible PDAC variants? (also in the future)
2) please provide histological images (with attention of providing also an acceptable resolution from the murine model)
3) Please expand the discussion on immunotherapy in PDAC, also taking into account ongoing trials that may benefit of this new model-based approaches
4) Figure 1: please provide/add MRI-images on day 8 and day 27
5) Figure 4: use different colors and not only this gray-scale
6) please discuss the findings also based on this seminal paper on PDAC microenvironment (PMID: 29661773)
Round 2
Reviewer 1 Report
Thank you to the authors for addressing my concerns. While the paper would be significantly improved with additional phenotyping of the immune cell subsets and studying the intratumoral environment, the extension on the discussion is appreciated.
Reviewer 2 Report
The authors addressed well the commenta and suggestions